# Effects of Shear Tabs and High-Strength Bolts in Seismic Performance of Steel Moment Connections

Chi-Ming Lai [1], Ching-Yu Yeh [2], Sin-Yu Kang [1] and Heui-Yung Chang [3],*

[1] Department of Civil Engineering, National Cheng Kung University, University Road, Tainan 701, Taiwan; cmlai@mail.ncku.edu.tw (C.-M.L.); summer199713@gmail.com (S.-Y.K.)
[2] Ruentex Engineering & Construction Co., Ltd., Bade Road, Taipei City 104, Taiwan; startagain98@gmail.com
[3] Department of Civil Engineering, National Chung Hsing University, 145, Xingda Road, Taichung 402, Taiwan
* Correspondence: hychang586@dragon.nchu.edu.tw

**Abstract:** A shear tab and high-strength bolts are often used to connect a steel H-beam to a column. The design demand and capacity of these elements vary from one standard to the other. To investigate the effect, this study applied a finite element method (FEM) to develop models for two steel moment connections and validated the effectiveness by test data. The connections were characteristic of bolted-web-and-welded-flange details. The FEM models were then used to study the design of shear tabs and high-strength bolts in accordance with the U.S. and Japan standards and compared to the Taiwan practice. The result showed a small difference in the peak loads of the connections. However, the U.S. direct welded flange connection had flange buckling and strength degradation at a relatively smaller drift. The connection had a thinner shear tab and fewer high-strength bolts. The other two connections had very similar design results and loading responses. The increase in shear-tab thickness reduced the stress concentration and fracture potential of the connections. It is, therefore, recommended to design a shear tab with moment capacity greater than the beam web. This will reduce the stress concentration of the base metal surrounding the beam-flange groove welds, increasing the connection ductility.

**Keywords:** finite element method (FEM); steel moment connections; shear tab; high-strength bolt; design practice and standards; moment capacity; connection ductility; severe seismic application

## 1. Introduction

Brittle fractures of steel moment connections were observed in the buildings attacked by the 1994 Northridge earthquake, and by the 1995 Kobe earthquake. The connections were characteristic of bolted-web-and-welded-flange (BWWF) details. The stress concentration was observed of the beam flange groove welds and the surrounding base metal. This raised concern about inadequate participation of the bolted web connections in transferring bending moments [1,2]. The test results were summarized of a post-Northridge steel moment connection design proposed by the SAC Joint Venture [3]. It was found that weld fracture mitigation measures may improve but cannot ensure the plastic deformation capacity of a steel moment connection against strong earthquake shaking. A 3D nonlinear finite element method (FEM) was used to further study the effects of local geometric details [4,5]. The results showed that shapes, sizes and alike geometric properties can also affect the fracture potential of the connections.

It did not attract much attention until quite recently to study the cyclic behavior of deep wide-flange sections and hollow structural sections (HSS), used as columns in steel special moment frames (SMFs) [6–12]. The results of FEM simulations and experimental evaluation demonstrated that wide-flange beam-columns with slenderness ratios near the AISC limits may experience rapid deterioration in flexural strength under high-ratio axial loads [6]. Moreover, the AISC axial load–bending moment interaction equation can underestimate by at least 30% the columns' ultimate strength [7]. For HSS columns, the AISC limits may be

conservative in the width-to-thickness ratios but may not be conservative in the depth-to-thickness ratios [8]. Most recently, efforts were made to study out-of-plane flexural capacity, initial stiffness and hysteretic models for circular and rectangular HSS X-connections [9–11]. The former investigation showed that the early global instability of deep columns can lead the adopted SMRFs to collapse at relatively small drifts [12].

While steel H-beam-to-H-column connections are commonly adopted in the U.S., steel H-beam-to-Box-column connections are more often used in Japan, Taiwan, and other regions of high seismicity. The FEM simulation was performed, and the result showed the difference in stress distribution between these two types of steel connections [13]. Several large-scale specimens, including two non-reinforced connections (i.e., typical steel moment connections with BWWF details) were tested, and the results showed that both the connections failed by the brittle fracture of the beam flanges, and the cracks initiated from the roots of the weld access holes [13,14]. The causes of the failure were also identified through 3D nonlinear FEM simulation. The stress concentration was found near the weld access hole, and that showed the potential causing the beam flange to fracture.

A series of member tests and shaking table tests were carried out to examine the retrofit schemes of adding supplemental welds, wing plates, and a haunch for the steel moment connections adopted in a 1970s Japanese high-rise steel building [15,16]. The result showed that adding welds between the shear tabs to the column faces can effectively reduce the strain concentration at the bottom flanges, greatly increasing the cumulative plastic deformation capacity of the steel connections. Four steel H-beam-to-box-column connections were tested under the 2005 AISC seismic provisions to confirm the effectiveness of rehabilitating steel moment connections by welding full-depth side plates between the column face and beam flange inner side [17]. Two of the connections were taken from an existing SMRF in Kaohsiung, Taiwan. The non-rehabilitated connection sustained a 3% drift and did not meet the AISC plastic rotation requirement. The other three rehabilitated connections sustained a drift greater than 4% without fractures. The side plates helped transfer the beam moments to the column, reducing the beam–flange tensile strain near the column face.

These studies not only show the advantage of adding welds or side plates to the column faces, but also illustrate a possibility of improving the moment transferring and connection ductility [15–17]. Shear tabs and high strength bolts help transfer the beam moments and can be expected to also affect the connection performance. To investigate the effect, two full-scale steel H-beam-to-box-column connections were designed and tested in the last phase of this study [18]. The shear tabs and high-strength bolts of the two connections were designed to completely transfer the moments from the beam webs to the columns. One connection tested the post-earthquake design practice adopted in the field of structural engineering in Taiwan. The perceptions about bolt slippage and the probably devastating effects have caused the structural engineers in Taiwan to improve the design practice. The bolted web connections were designed using slip strength of high-strength bolts and considering the eccentricity moment. The other connection examined a new retrofit scheme of adding another shear tab and high-strength bolts. Both the connections were tested under the 2016 AISC seismic condition [19] and sustained a drift of 4%.

This paper presents the results of FEM simulation and parametric study on steel H-beam-to-box-column connections with BWWF details. In detail, FEM models were first developed for the two steel connections and then validated by the results of seismic tests [18]. Dedicated efforts were taken to simulate the BWWF details. Special attention was paid to the behavior of high-strength bolts, including bolt pretension, slip, contact and bearing. In the last phase of this study, the slip behavior of a high-strength bolt and the pretension loss was modeled and validated by a component test [20]. The FEM models were used to further study the design demand and capacity of shear-tabs and high-strength bolts in accordance with the U.S. and Japan standards and compared to the Taiwan practice. The design parameters included (1) bolt arrangement and strength, (2) shapes of weld access

holes, and (3) shear-tab thickness. The results obtained will help investigate the role of shear tabs and high-strength bolts in the seismic performance of steel moment connections.

## 2. Materials and Methods

### 2.1. FEM Simulation

The techniques of FEM simulation were applied to develop models for steel moment connections with BWWF details. The simulation was made using the commercial software ANSYS Workbench (version 16.2 and 17.0) [21].

### 2.2. Assumptions and Premises

The simulation was made under the following assumptions and premises:

(1)   The FEM simulation did not include (a) welding residual stress and thermal effects, or (b) tensile fracture of steel plates, bolt and weld materials.
(2)   Full-scale models were established for steel moment connections with BWWF details.

Efforts were taken to model in detail the shear tab, high-strength bolts and welds. This allowed further studying of the beam moments transferring and the effect of bolt slippage.

These assumptions and premises help the FEM simulation achieve high accuracy and efficiency.

### 2.3. Boundary Conditions and Meshing

Figure 1a gives an example showing the boundary conditions and FEM meshing. The FEM simulation had the same specimens and boundary conditions as the connection tests [15]. In detail, the box column was fixed at the two ends. A shear tab was welded to the column face and connected the column to the beam web together with high-strength bolts. Complete joint penetration (CJP) welds were then made to connect the beam flanges and column. A displacement or an external load was applied at the tip of the beam. The beam was laterally braced near the one- and two-thirds points. The bracing prevented the beam from lateral torsional buckling. Figure 1b shows the FEM modeling of shear tabs, high-strength bolts and welds. These connecting elements were modeled, using dense meshes and contact elements (see Section 2.6 for more details).

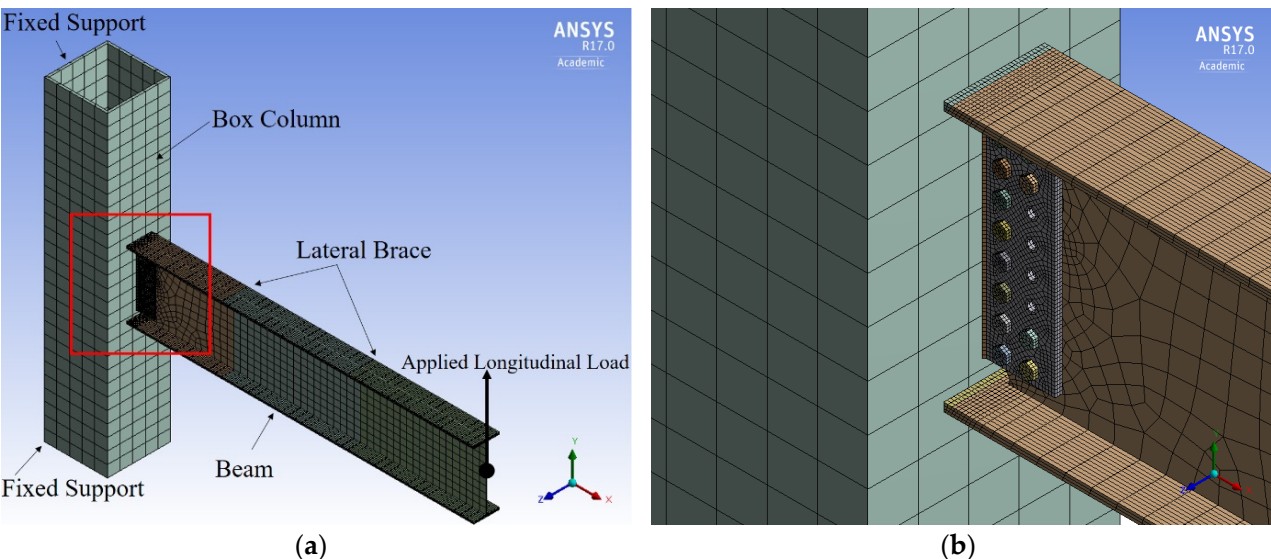

**Figure 1.** FEM model of a full-scale steel H-beam-to-box-column connection; (**a**) boundary conditions and mesh, (**b**) BWWF details.

## 2.4. Sections and Materials

Table 1 summarizes the details of the bolted web connections and mechanical properties of steel material. The Box 750 × 750 × 28 (mm) column and H-800 × 300 × 14 × 25 (mm) beam were composed of SN490B steel plates. The yield-stress-to-tensile-strength ratio $F_y/F_u$ was equal to or smaller than 80% for the SN490B steel. A piece of SN490B-steel plate with thickness of 22 mm was used as a shear tab in the UN-connection, which reflected the post-earthquake design practice in the field of structural engineering in Taiwan. This unreinforced connection was designated as "UN-connection". A pair of A572 Gr. 50 steel plates with thickness of 16 mm were used as shear tabs in the DS-connection, which examined a retrofit scheme for the pre-Northridge connection. The connection was reinforced, using double shear tabs, and was designated as "DS-connection". FE70 weld material was used. The table also gives the strength properties of JIS S10T high-strength bolts and F14T super high-strength bolts.

**Table 1.** Details of bolted web connections and mechanical properties of steel material.

| 1. Details of bolted web connections | | |
|---|---|---|
| | **Shear tab(s)** | **High-strength bolts** |
| UN-connection | SN490B steel, PL-640 × 130 × 22 | 12-M24 S10T |
| DB-connection | A572 Gr. 50 steel, 2PL-640 × 130 × 16 | 12-M22 F14T |

| 2. Mechanical properties of steel material | | | |
|---|---|---|---|
| **Steel material** | **$F_y$ (MPa)** | **$F_u$ (MPa)** | **Elongation (%)** |
| SN490B steel | 325~445 | 490~610 | Flange ≥ 21 Web ≥ 17 |
| A572 Gr. 50 steel | ≥345 | ≥450 | ≥18 |
| $F_{E70}$ weld material | 555 | 592 | ≥26 |
| JIS S10T bolt | 920 | 1000~1200 | ≥14 |
| JIS F14T bolt | 1260 | 1400~1490 | ≥14 |

| 3. Strength properties of high-strength bolts | | | |
|---|---|---|---|
| **High-strength bolt** | **Pretension (kN)** | **Slip strength [1] (kN)** | **Shear strength [2] (kN)** |
| M24-S10T | 233 | 118 | 166 |
| M22-F14T | 330 | 336 | 426 |

[1] Slip coefficient = 0.45 for sand-sprayed surfaces; [2] S10T and F14T high-strength bolts are used respectively in single and double shear.

## 2.5. Stress–Strain Relation

The elastic and plastic behaviors of steel plates, weld materials and high-strength bolts were simulated, using an isotropic model and a kinematics model, respectively. The stress–strain curve was simulated, using a trilinear model. Table 2 gives the yield strength, tensile strength and strain hardening rate ($E_{sh}$). The strength parameters were originally obtained from material tests and then calibrated for the FEM simulation of connection tests [15]. In detail, Young's modulus ($E$) and Poisson's ratio ($v$) were 200 GPa and 0.3, respectively. For the SN490 steel plates, for example, the stress–strain relation was initially simulated with a yield strength of 373 MPa, using a bilinear relation with a strain hardening rate $E_{sh}$ of 0.04. After the stress increased to the tensile strength of 528 MPa, the stress was kept the same.

**Table 2.** Strength parameters of trilinear models.

| Steel Material | A572 Gr. 50 | SN490B | $F_{E70}$ | S10T | F14T |
|---|---|---|---|---|---|
| Yield strength (MPa) | 350 | 373 | 555 | 920 | 1260 |
| Tensile strength (MPa) | 500 | 528 | 592 | 1000 | 1400 |
| Strain hardening exponent ($E_{sh}$) | 0.04E | 0.04E | 0.0036E | 0.05E | 0.05E |

### 2.6. Elements and Interfaces

Fabricated, welded steel H-beams and box-columns were used in this study. Inside the steel box column, two continuity plates were welded at the same heights with the top and bottom beam flanges so as to enhance the moment transferring between the beam and column. The regular shapes, such as beam flanges, column plates, continuity plates and welds, were modeled with 20-noded solid hexahedral elements (SOLID186), whereas the irregular shapes, such as beam webs with welded access holes, shear tabs with bolt holes, and high-strength bolts, were modeled with 10-noded solid tetrahexon elements (SOLID187). To ensure the computational efficiency, most of the elements had a ratio of length to width of about 1.0, and the maximum was up to 20. The beam flanges, column plates and continuity plates had rectangular meshes, which were connected at their common interface with a bonded-always, contact and target pair. Accordingly, the bolts were bonded with the nuts in the same way.

The studied steel beam-to-column connections had welded joints and bolted joints, namely, fillet welds connected a shear tab to the column. CJP groove welds were made between the beam flanges and column. On the other hand, high-strength bolts were used to connect the shear tab to the beam web. The behaviors of the welded joints were simulated, using surface-to-surface contact elements (CONTA177). The simulation considered the contact and sliding between the welded surfaces and deformable line segments. As for the bolted joint, the shear tab and high-strength bolts were connected with a frictional interface. The beam web and bolt nuts were connected in the same way. The meshes of the shear tab and beam web were connected at their common interface with a frictional-always, contact and target pair.

### 2.7. Modeling of Bolt Slippage

In the last phase of this study, dedicated efforts were taken to develop a physical model to simulate the behavior of a high-strength bolt, including bolt pretension, slip, contact and bearing [20]. A series of lap connection testing, as shown in Figure 2a, was carried out, and the results were used to directly validate the FEM model of bolt slippage. Compared to the steel beam-to-column connections mentioned above, lap connections are considered to be more appropriate to study the slip behavior of high strength bolts. This is especially so because the lap connections are much easier to measure the slip deformations and resistance.

In the FEM modeling, as shown in Figure 2b, the steel plates and high-strengths bolt were connected with a frictional surface. The coefficient of friction was the same as the slip coefficient of high-strength bolts: 0.45 for a blasted clean surface [22]. The slip coefficient was also validated by a full-scale steel bolted-web connection [18]. Bolt holes were assumed to be 1.5 mm larger than the bolt diameter. The AIJ (2012) [22]–specified minimum bolt pretension forces were generated via a "pretension element" (PREST 179).

In this study, the bolt threads were not modeled, and the bolt nut was glued to the bolt shank. Similar approaches to modeling the pretension of high strength bolts were adopted in previous studies (e.g., [23,24]). That allowed simulating the tightening of a high-strength bolt by reducing the effective length (length between the underside of the bolt head and nut). That also allowed simulating the loss of bolt pretension by increasing the effective length. The presented study locked the remaining pretension until the high-strength bolt slipped again.

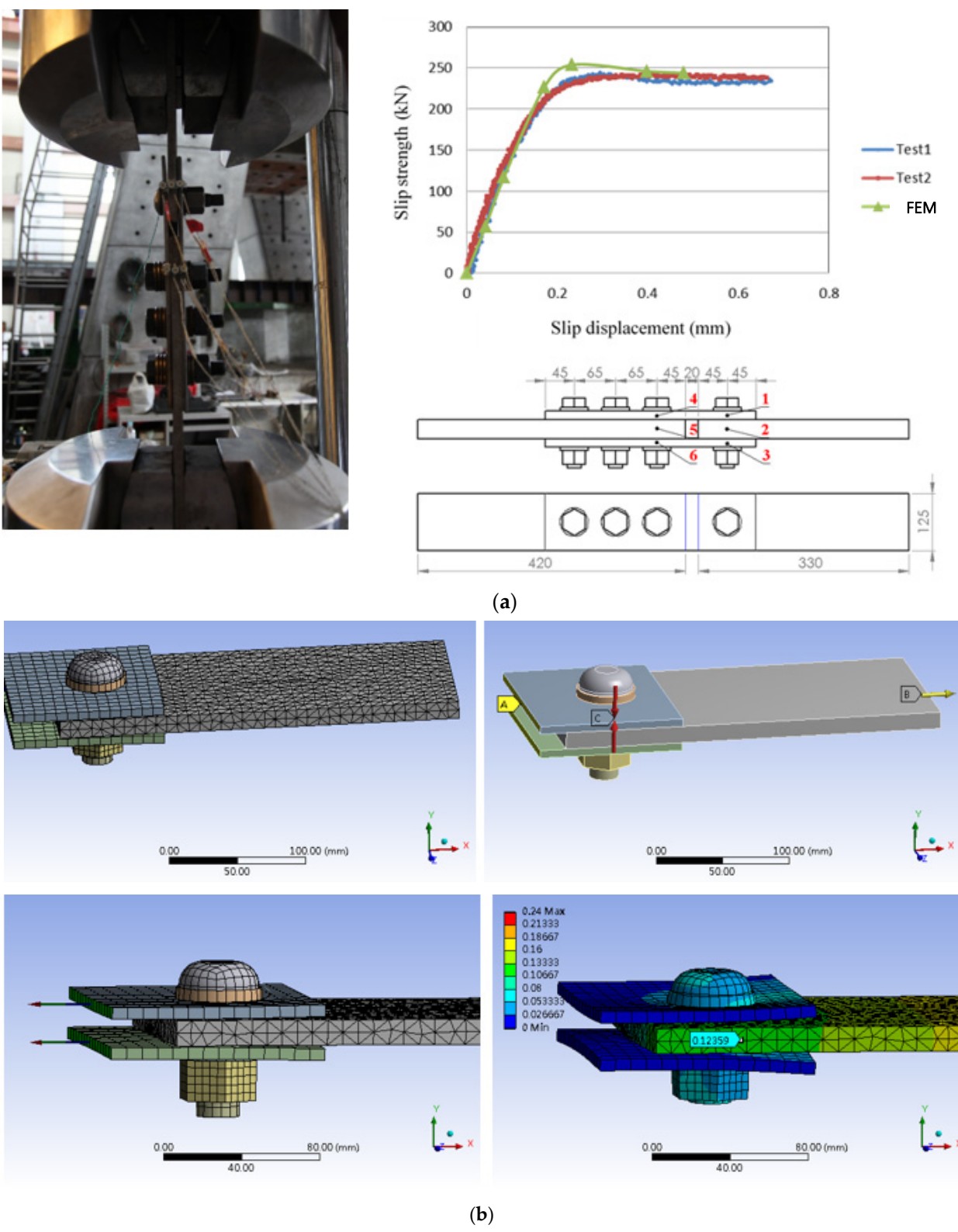

**Figure 2.** FEM simulation of bolt slippage; (**a**) model validation, (**b**) FEM simulation.

*2.8. von Mises Stress and PEEQ Index*

The von Mises stress is often used to predict the yielding of materials under multiaxial loading [25]. With the principal stresses, $\sigma_1$, $\sigma_2$, and $\sigma_3$, one may calculate the von Mises stress using the following equation:

$$\sigma_\theta = \sqrt{\frac{1}{2}[(\sigma_1 - \sigma_2)^2 + (\sigma_2 - \sigma_3)^2 + (\sigma_3 - \sigma_1)^2]} \tag{1}$$

The *PEEQ* index is used to evaluate the demands of plastic deformations and to show the potential of fracture failures [4]. With the vector component of plastic strain $\varepsilon_{ij}$ in the direction of *i* and *j*, one may calculate the *PEEQ* index using the following equation:

$$PEEQ = \sqrt{\frac{2}{3}\varepsilon_{ij}\varepsilon_{ij}} \tag{2}$$

### 3. FEM Simulation of Connection Tests

*3.1. Validation by Connection Tests*

Load-Displacement Responses

Figure 3a,b shows the predicted load-displacement responses for the two steel connections and compares them to the experimental responses. For reference, the AISC loading protocol is depicted as well. The connection models were validated by the results of two full-scale tests [18]. Displacements were applied 4.125 m from the column face at the beam tip under the 2016 AISC seismic condition [19].

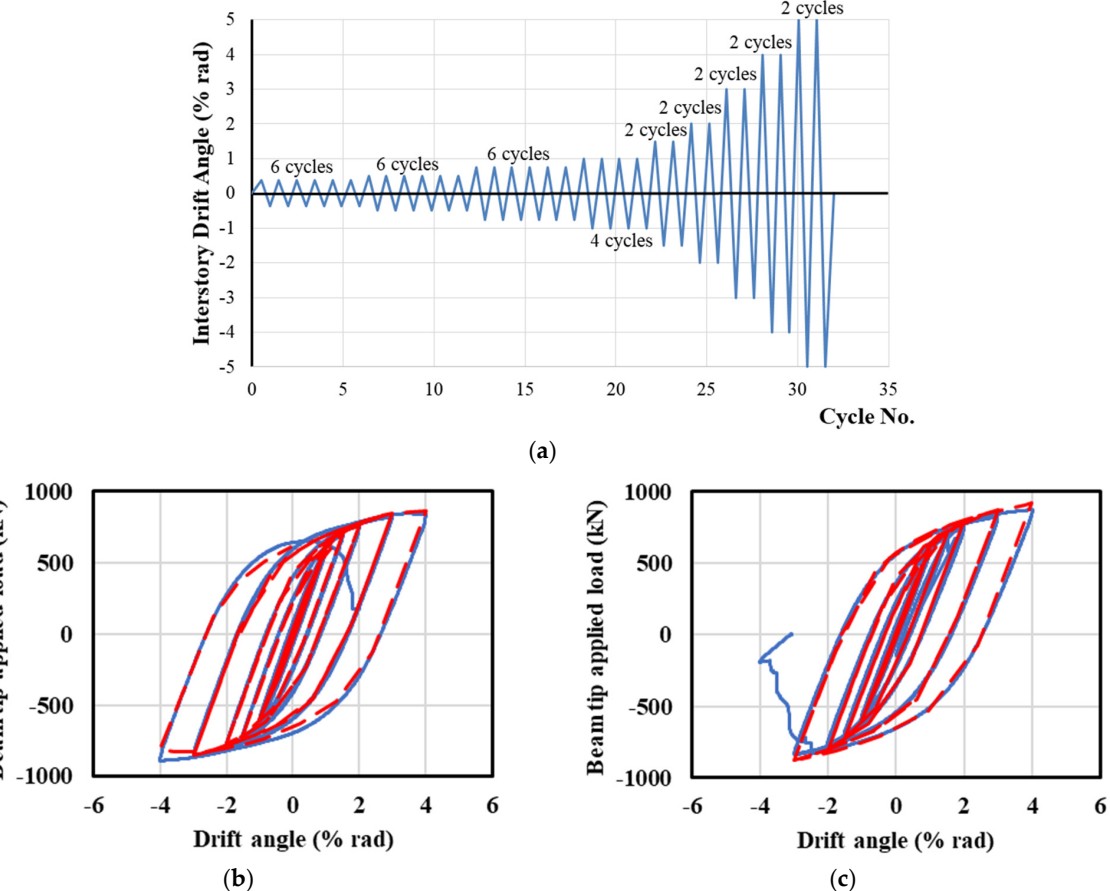

**Figure 3.** Loading protocol and simulation of connection tests: (**a**) AISC loading protocol, (**b**) UR-connection, (**c**) DS-connection (simulation in red and test in blue).

As mentioned, the UN-connection reflected the post-earthquake design practice in the field of structural engineering in Taiwan. The DS-connection examined a retrofit scheme of adding another shear tab and high-strength bolts to the pre-Northridge connection. The post-earthquake design code (such as AISC 2016 [19]) requires the drift of 4% at least. Both the connections were tested and sustained to a drift of 4%.

The results of the FEM simulation can be considered to agree well with the test results. This is partly attributed to the detailed modeling of shear tabs, high-strength bolts and welds. The results of FEM simulation were also used to find the root cause of fracture at fillet welds between shear tabs and column face for the DS-connection [18]. This suggests that instead of yield strength, tensile strength of steel plates should be used to evaluate the strength demands of the fillet welds. For the limit of space, only the responses of the UR-connection are presented and discussed in detail in the following.

### 3.2. von Mises Stress and Connection Behavior

Figure 4 shows the distribution of von Mises stress and compares it to the experimental results. As illustrated by Figure 4a, the yielding of the steel plates caused paint to peel off the beam flanges, and the web of the UR-connection was subjected to a 2% drift in the experiment. As depicted in the same figure, the simulated stress values show good agreement with the experimental observation. The SN490 steel plates was simulated with a yield strength of 373 MPa, using a bilinear relation with a strain hardening rate $E_{sh}$ of 0.04. After the stress increased to the tensile strength of 528 MPa, the stress was kept the same. As can be seen in Figure 4b, by the end of the experiment, the UR-connection sustained a 4% drift, and local plate buckling occurred in the bottom flange of the connection. As also can be seen there, the FEM simulation also predicts the connection to have local plate buckling in the bottom flange.

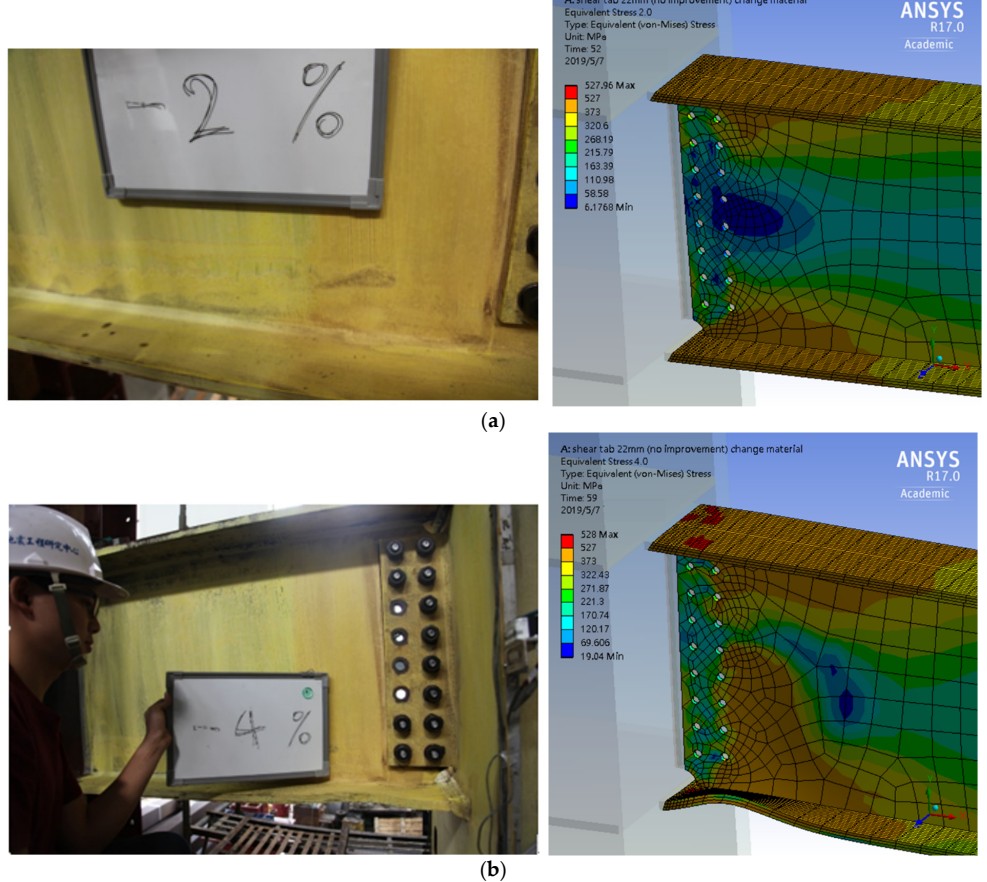

(a)

(b)

**Figure 4.** UR-connection behavior and von Mises stress distribution: (**a**) 2% drift, (**b**) 4% drift.

### 3.3. Strain Measurement

Figure 5 depicts the measured and simulated strain for different drifts. The strain gauges F260 and W260 were attached on the beam flanges and were 260 mm away from the column face. As illustrated by Figure 5a, the uniaxial strain gauges F260 were attached symmetrically from the central line of the beam flange. As can be seen in Figure 5b, the triaxial stain gauges W260 were attached outsides the shear tab and in the same lines of high-strength bolts.

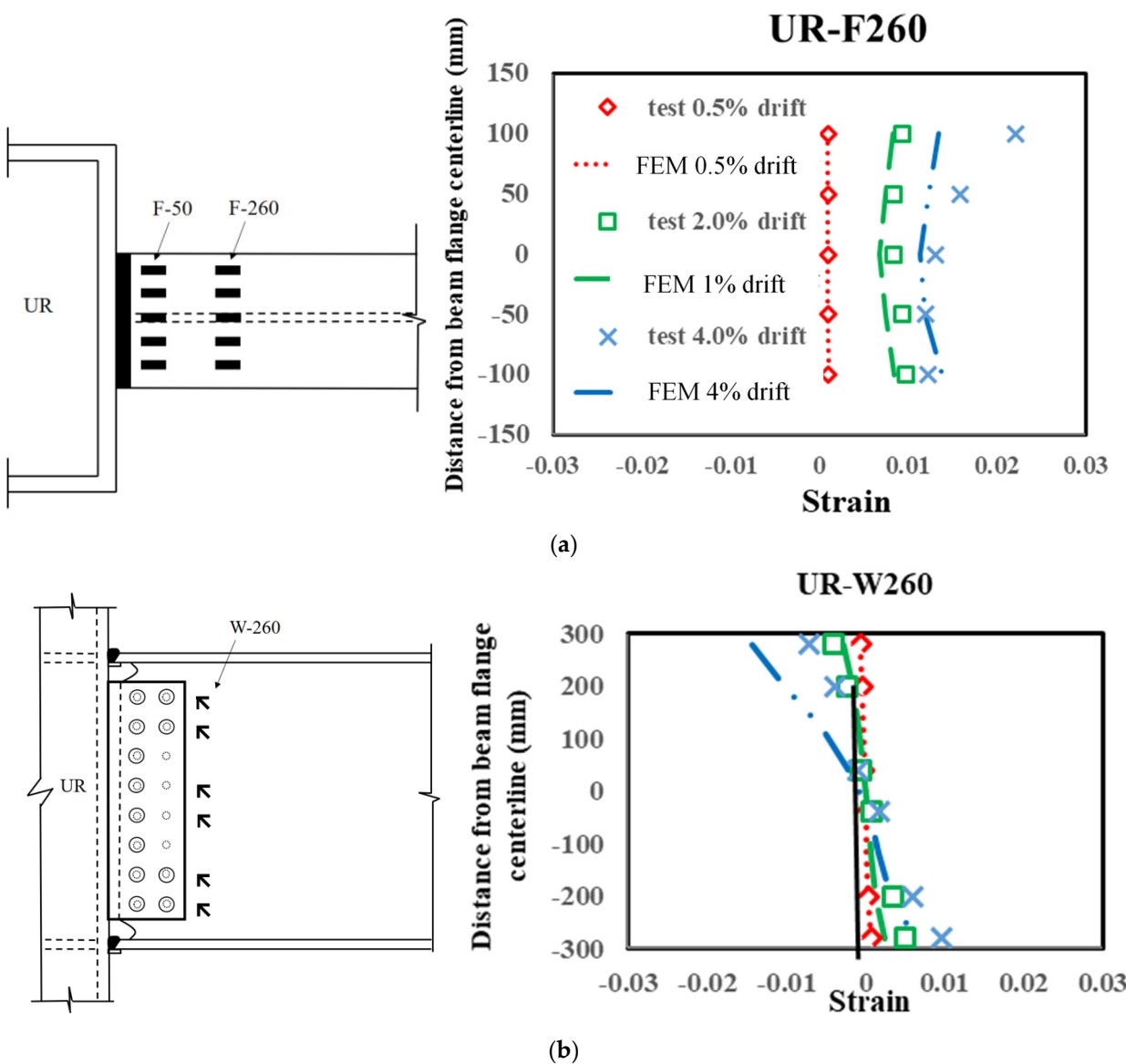

**Figure 5.** FEM simulation of strain measurement: (**a**) flexural strain, (**b**) shear strain.

For the 0.5% drift, the FEM simulation gave a very good estimation of the elastic strain values. As the drift increased to 2%, the FEM simulation slightly underestimated the plastic strain values. The simulation can still be considered to give a good approximation of the measurement. As the drift increased to 4%, the FEM simulation underestimated the plastic strain values to a greater extent. In detail, local plate buckling caused the strain distribution to lose the symmetry. The effects varied significantly depending on the strain types and gauge locations. Moreover, the effects were more noticeable in the test than in the simulation. The stress–strain distribution near the column face is very complicated. For computational effectiveness and efficiency, the stress–strain relation was simulated using a

trilinear model, and the strength parameters were validated by connection tests and FEM simulation. The effects were ignored of welding heat and residual stress. Such factors are also affected, and the effect needs further evaluating. Despite that, the FEM simulation can still be considered to give reasonably good estimates of the results of the connection tests, especially the load-displacement curves and fracture patterns.

## 4. Parametric Analysis and Cases Study

### 4.1. Case Study

4.1.1. Design Consideration and Examples

Figure 6 explains the design consideration of bolted web connection in the U.S., Taiwan and Japan. For the ease of fabrication, the U.S. direct welded flange connection adopts a shear tab or web plate and high-strength bolts in Figure 6a. The beam flanges are designed to carry all the beam moment, and the shear tab is designed to transfer shear without considering any eccentricity moment [26]. The shear strength of high-strength bolts were used in the strength evaluation. The development of structural design standards and seismic provisions in Taiwan basically follow those of the U.S. high-strength bolts and were observed to slip early in the cyclic loading tests of steel moment connections with BWWF details (e.g., [2,24]). The perceptions about bolt slippage and the probably devastating effects [1,2,27] have caused the structural engineers in Taiwan to improve the design practice. The bolted web connections were designed, using the slip strength of high-strength bolts and considering the eccentricity moment in Figure 6b. In Japan, a beam web is designed to take moment with the beam flanges. The 2012 AIJ connection design recommendations [22] suggest that the high-strength bolts of a bolted web connection be designed to take the shear and moment in Figure 6c. The shear strength of high-strength bolts were used in the strength evaluation.

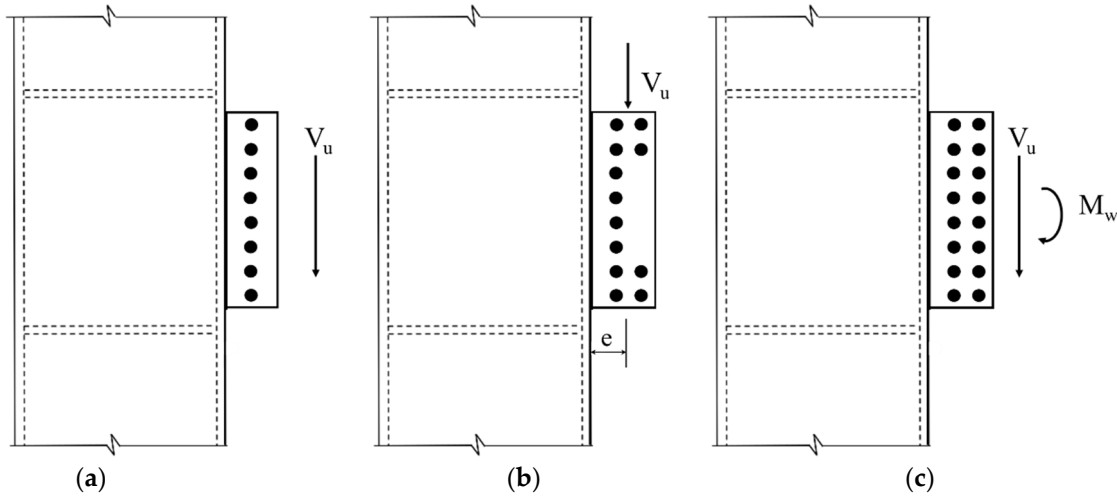

**Figure 6.** Design consideration of bolted web connections: (**a**) the U.S., (**b**) Taiwan, (**c**) Japan.

In sum, the UR-connection uses a total of 12 M24-S10T high-strength bolts with a 22 mm thick shear tab (denoted as PL22B12). If the moment strength of the beam web were to be taken as the design demand, the shear tab would have a plate thickness of more than 19 mm. The U.S. direct welded connection uses a row of 8 M24-S10T high-strength bolts with a 12 mm thick shear tab (denoted as PL12B8). The beam web has a plate thickness of 14 mm and length of 750 mm. The shear tab of the connection has a thickness of 12 mm with the length of 640 mm. The shear-tab thickness has varied with the design demand. If a shear tab or web plate were to be designed based on the shear demand on the beam web, the web plate would be thinner than the beam web. The Japan connection has two rows of 8 M24-S10T high-strength bolts with a 22 mm thick shear tab (denoted as PL22B16).

### 4.1.2. Evaluation of Connection Performance

In the company paper [18], the results were presented of two full-scale connection tests. The UR-connection was used to test the post-earthquake design practice adopted in the field of structural engineering in Taiwan. The 22 mm thick shear tab ensured the moment transferring between the beam web and column face. The DS-connection was used to examine a new retrofit scheme. If the design demand were to be evaluated, using the shear strength of the beam web, the plate thickness of the shear tab would be one size up, in comparison to that of a beam web. The beam web has a plate thickness of 14 mm. In this case, the shear tab is required to have a plate thickness of more than 16 mm. A 16 mm thick shear tab was considered unable to completely transfer moment, and the connection was then retrofitted by adding another shear tab. Both the connections were tested under the 2016 AISC seismic condition and had fracture in base metal surrounding the beam flange groove welds after sustaining a maximum 4% drift.

The FEM simulation allowed further evaluating of the effects of design methodologies and practice, as shown in Figure 7.

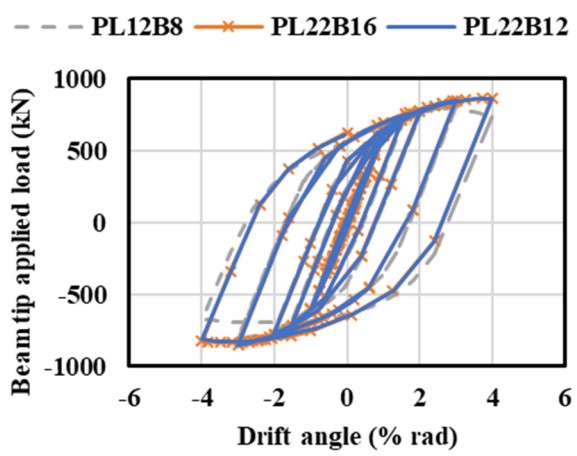

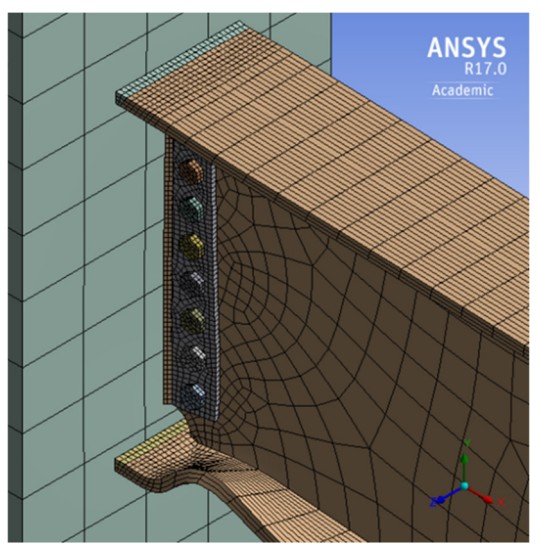

**(b)**

| Model | PL12B8 (US) | PL22B16 (JP) | PL22B12 (UR) |
|---|---|---|---|
| $F_{max}$ (kN) | +814.25 | +861.2 | +862.63 |
| | -816.28 | -853.32 | -852.66 |
| $F_{4\%}$ (kN) | +737.87 | +861.2 | +862.63 |
| | -673.63 | -824.75 | -813.58 |

**(a)**

**Figure 7.** FEM simulation of design examples: (**a**) load-displacement curves, and (**b**) local buckling of bottom beam flange (PL12B8 model).

The load-displacement curves of the three connections were simulated and compared as shown in Figure 7a. The peak loads ($F_{max}$) and loads at the 4% drift ($F_{4\%}$) are also given there. For the peak loads ($F_{max}$), the ratio between the U.S. and U.R. connections was 94.62% (=816.28/862.63). In other words, the connection strength almost did not vary with the design methodologies and practice. For the 4% drift load ($F_{max}$), the ratio changed to 85.53% (=737.87/862.63). The U.S. connection developed the maximum strength at a drift smaller than 4%. After that, local plate buckling occurred in the bottom beam flange, as shown in Figure 7b. The buckling has caused more strength degradation in the U.S. connection

(i.e., 737.87/816.28 = 90.39%) than the U.R. connection (i.e., 813.58/862.63 = 94.31%). On the other hand, the UR and JP connections had very similar strength values.

### 4.2. Parametric Study

#### 4.2.1. Response to Monotonic and Cyclic Loadings

Figure 8 depicts and compares the responses to monotonic and cyclic loadings. As mentioned, the UR-connection was tested under the 2016 AISC seismic condition, and the fracture of base metal surrounding the beam flange groove weld occurred after the connection sustained a drift of 4%. The force-displacement curves were simulated for monotonically and cyclic loadings in Figure 8a. The max values of PEEQ index were calculated for the base metal surrounding the beam flange groove weld in Figure 8b. The PEEQ index was used to evaluate the demands of plastic deformations and to show the potential of fracture failures [4]. The simulation applied a displacement at the beam tip. The AISC loading protocol, as shown in Figure 3, was used to generate the cyclic loading response. For comparison, the simulation increased the displacement gradually to 4% drift and generated the monotonic loading response. The monotonic loading response gave a good envelop to the cyclic loading response. For the ease of comparing, only the monotonic loading responses were further simulated and are compared in detail in the following.

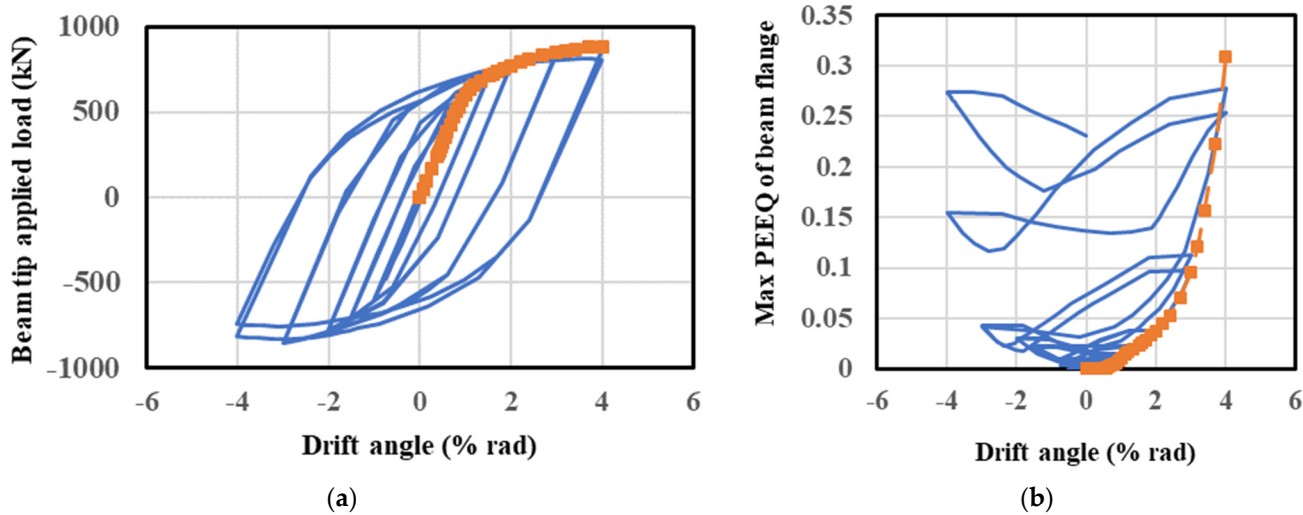

**Figure 8.** Response to monotonic loadings (in orange) and cyclic loadings (in blue): (**a**) force-displacement curves of UR-connection, and (**b**) max PEEQ index of base metal surrounding the beam flange groove weld.

#### 4.2.2. Effects of Bolt Arrangement and Strength

Figure 9 depicts and compares the values of PEEQ index for three different bolt arrangements and strength. There are three cases in the FEM simulation, as shown in Figure 9a. As mentioned, the UR-connection was designed in accordance with the post-earthquake design practice in Taiwan. The connection uses a total of 12 M24-S10T high-strength bolts with a 22-mm thick shear tab. This 12-bolts case took into account the bolt slip strength, eccentric moment and shear demand of the bolted web connection. The other two cases considered the shear strength of high strength bolts. In detail, the 8-bolts case considered the shear demand only, and the 16-bolts case considered both the shear and moment demands.

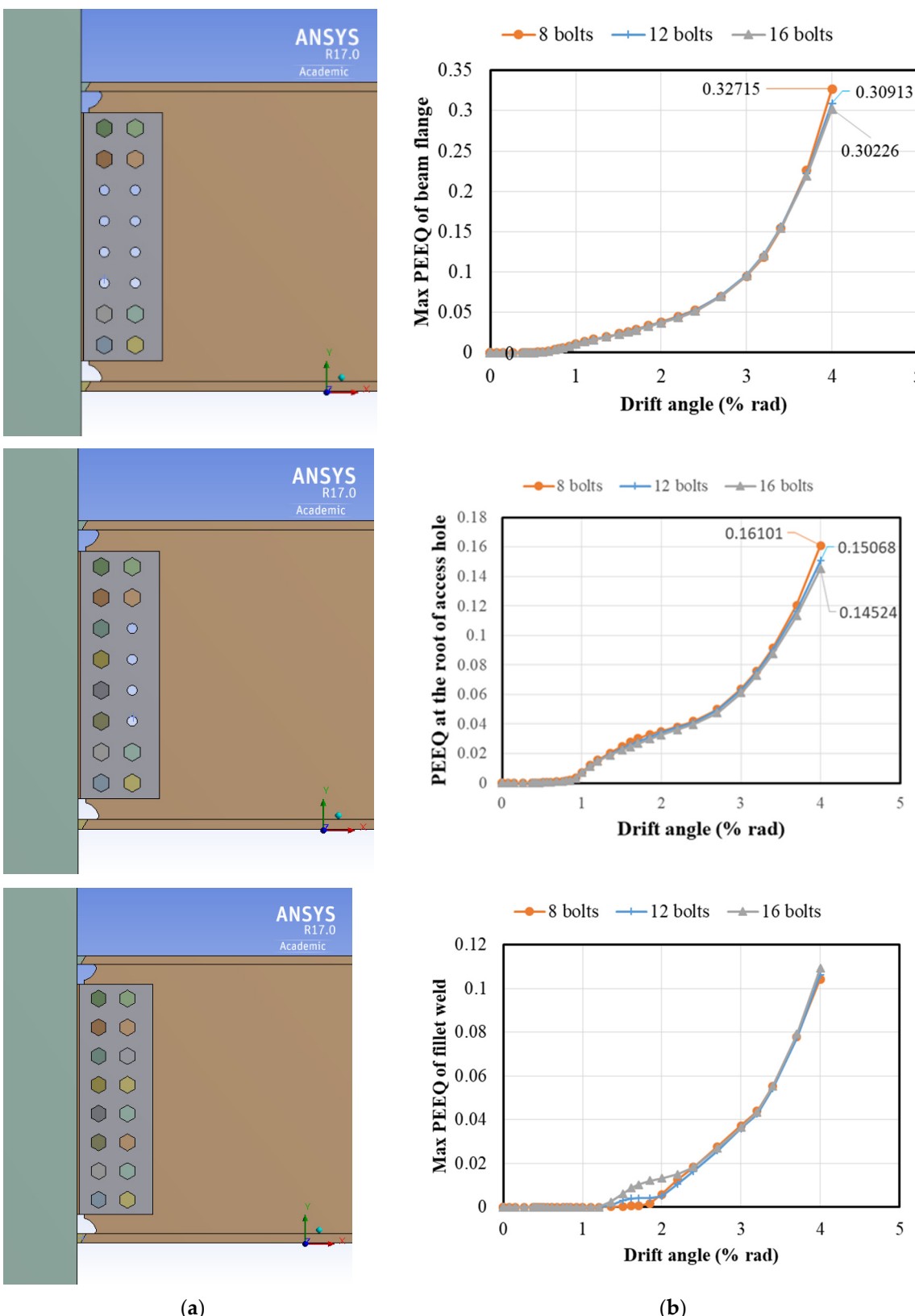

**Figure 9.** Effects of bolt arrangement and strength: (**a**) FEM model, and (**b**) PEEQ index (for a steel connection with a 22-mm thick shear tab and M22 S10T high-strength bolts).

The values of PEEQ index were compared in detail in three regions, as shown in Figure 9b. In detail, the max values of PEEQ index were compared in detail in the base

metal surrounding the beam flange groove weld, and in the fillet weld between the shear tab and column face, respectively. The values of PEEQ index were also compared at the root of the weld access hole. At a drift greater than 1%, the max PEEQ index in the fillet welds between shear tab and column face differed from one case to the other. As the drift increased to 2% and more, the PEEQ index became closer and then almost the same in the three cases. The small difference implies that the difference in bolt arrangements and strength can affect the connection behavior, but to a limited extent.

4.2.3. Shapes of Weld Access Holes and the Effects

Figure 10 shows the shapes of the weld access holes. For the ease of welding, a 1/4-circle cut was created with a radius ($r$) at the corner of a beam web, as shown in Figure 10a. Such a conventional weld access hole was thought to cause stress concentration, leading to fracture failures of the steel moment connections with BWWF details. After the earthquakes, special seismic weld access holes were developed with requirements in size, shape, and finish [22,28]. As illustrated by Figure 10b,c, the improved weld access holes provide a smoother transition to reduce the stress concentration, and the larger openings make it easier to access and inspect welds.

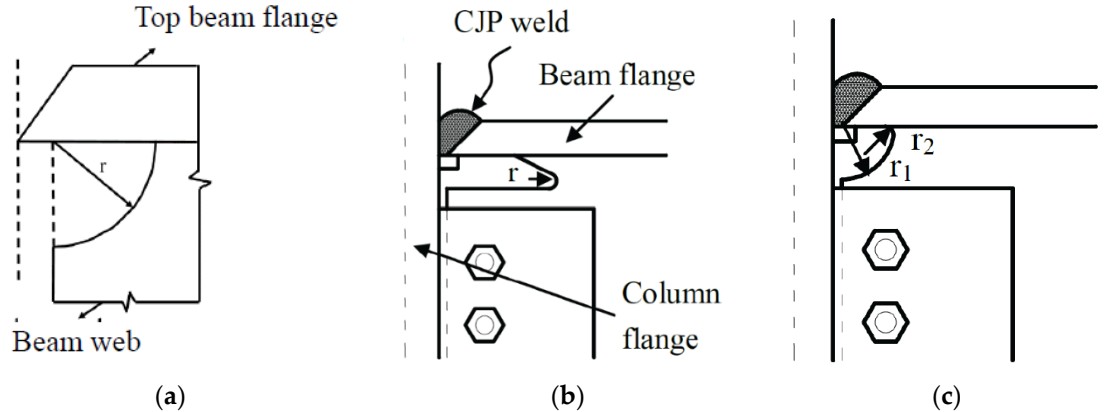

(a)    (b)    (c)

**Figure 10.** Shapes of weld access holes: (**a**) conventional type (e.g., $r$ = 35 mm, denoted as 35R), (**b**) FEMA-350 improved type, (**c**) JAS improved type I (e.g., $r_1$ = 35 mm, $r_2$ = 10 mm, denoted as 35R + 10R).

Figure 11 depicts and compares the values of the PEEQ index for conventional and improved weld access holes. Under the 2016 AISC seismic condition, the UR-connection was tested and sustained a drift of 4% before ductile fracture occurred in base metal surrounding the beam-flange groove weld. The improved weld access holes of the tested connections were created using two different radii (i.e., $r_1$ = 45 mm and $r_2$ = 10 mm). In the FEM simulation, the weld access holes of the connection were changed to be conventional ones with a radius (i.e., $r$ = 45 mm). The two types of weld access holes gave a very similar distribution of von Mises stresses in Figure 11a. The von Mises stresses near the weld access holes were analyzed and compared at the 4% drift. In detail, the stress at the root of the access hole did not reduce but increased for the improved weld access hole. The stress concentration in the region near the access hole also did not reduce. The two types of weld access holes also gave very similar distribution of the PEEQ index values in Figure 11b. The values of the PEEQ index near the weld access holes were also compared at the 4% drift. The PEEQ indices had the maximum at the root of the conventional weld access hole, and the peak point shifted for the improved access hole. Overall, the shapes of the weld access holes affected the distribution of the von Mises stress and PEEQ index values, but to a small extent.

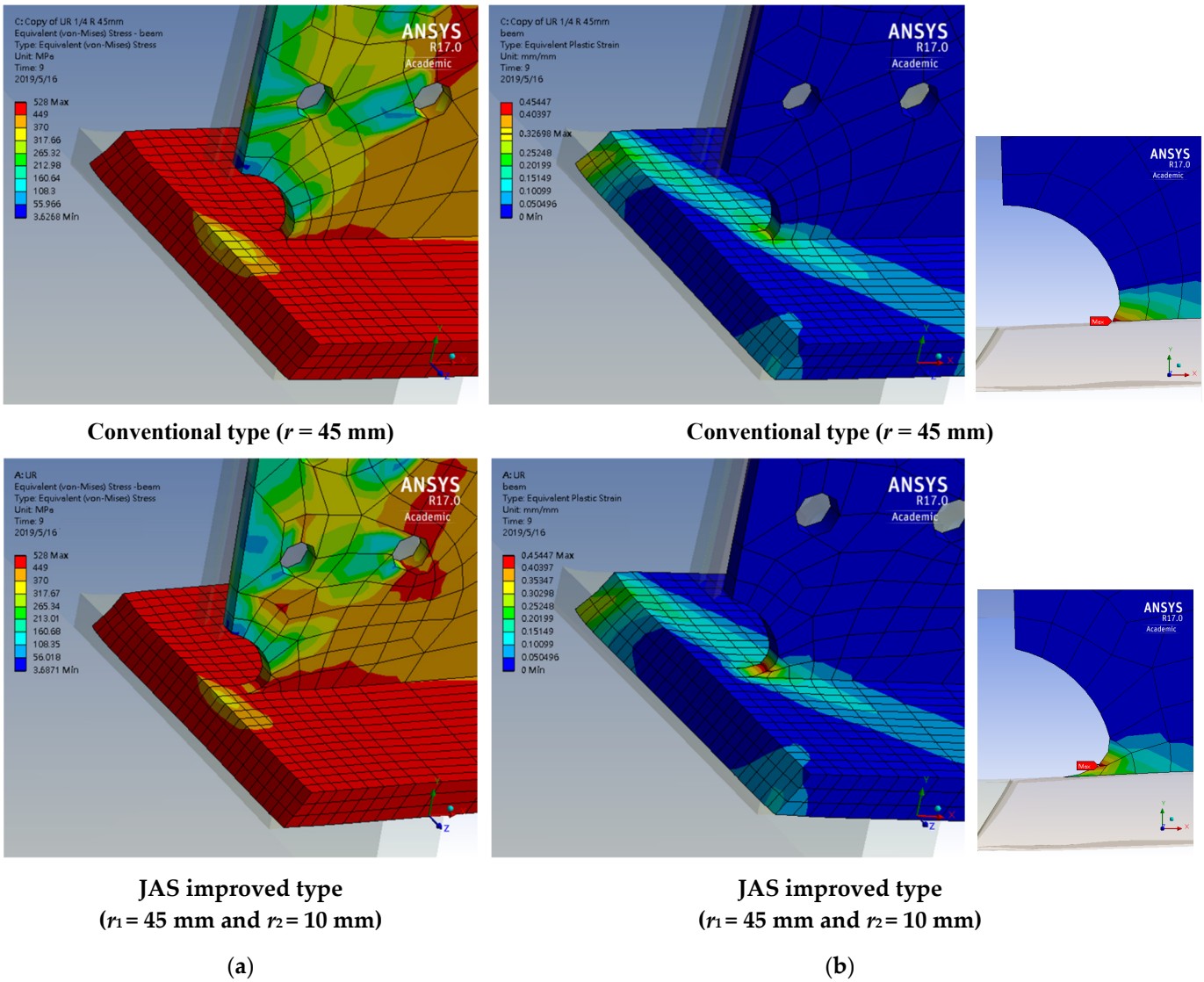

**Figure 11.** Effects of weld access holes: (**a**) von Mises stress, (**b**) max PEEQ index (drift = 4%).

Steel connections with BWWF details have geometric discontinuity in the weld access holes. The nature of geometric discontinuity causes stress concentration near the weld access holes. The effects make the connections unable to develop more plastic deformation capacity, like fully welded connections (e.g., [29,30]). The FEM simulation make it clearer that the shapes of weld access holes affect the distribution of von Mises stress and PEEQ index, but to a limited extent. The results have shown good agreement with previous studies [13,14]. In brief, improved weld access holes and the alike fracture mitigation measures are probably beneficial, but the effects are not sufficient to enhance the seismic performance of existing steel connections to the post-earthquake required level.

### 4.2.4. Shear-Tab Thickness and the Effects

Figure 12 depicts and compares the max values of PEEQ index for four kinds of shear-tab thickness. The strength of the fillet welds between the shear tab and column face was evaluated in accordance with the 2012 AIJ steel connection design recommendations [22]. The shear-tab thickness was 12, 16, 18, and 22 mm. The leg sizes of fillet welds were 8, 10, 12, and 14 mm. As mentioned, the UR-connection used a total of 12 M24-S10T high-strength bolts with a 22 mm thick shear tab. In this case, the moment strength of the beam web was taken as the design demand of the shear tab. The 18 mm case considered the moment demand on the beam web. The beam web and flanges took moment in proportion to the

elasticity section modulus. The 12 and 16 mm cases mainly considered shear demands. The former represented a direct weld connection and considered the shear demand on the beam. The latter took the shear strength of the beam web as the design demand of the shear tab.

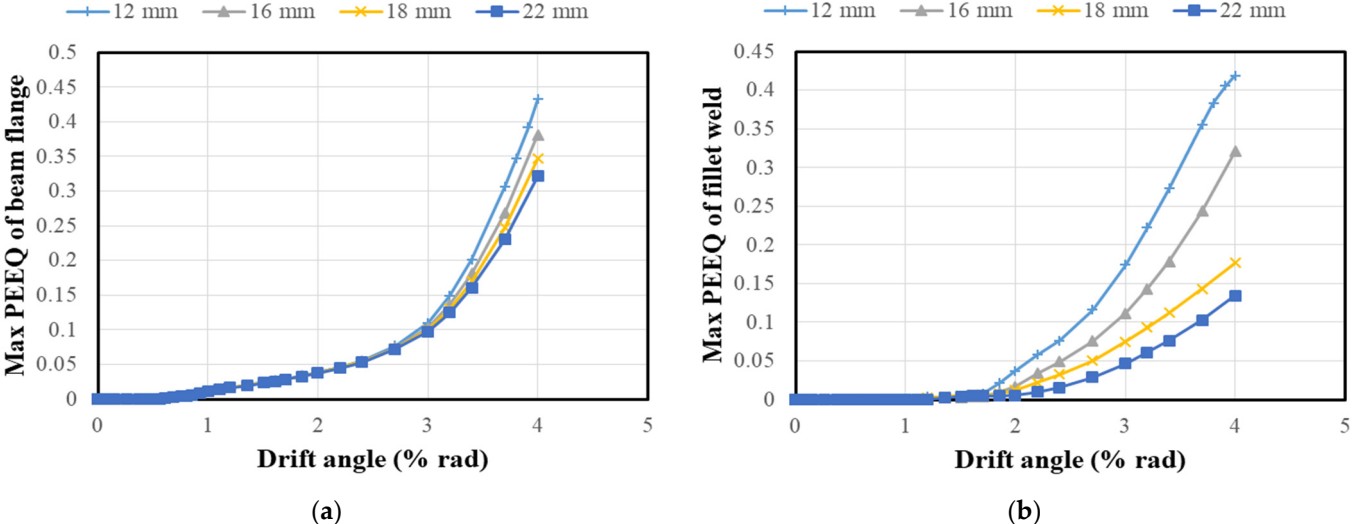

**Figure 12.** Shear-tab thickness and the effects on the max values of PEEQ index: (**a**) in base metal surrounding the beam flange groove weld, (**b**) in fillet weld between the shear tab and column face (of a steel connection with 12 M22 S10T high-strength bolts).

The max values of PEEQ index were calculated for the base metal surrounding the beam flange groove welds in Figure 12a. The values increased from zero after the plate yielded at a drift slightly greater than 1%. Moreover, the shear-tab thickness did not make a difference in the yield drift. After the drift increased to 3% and more, the values varied very differently depending on the shear-tab thickness. The thinner the shear tab, the greater the value. That has implied a higher potential of connection fracture, especially the one with a thin shear tab. The max values of PEEQ index were also analyzed for the fillet welds between shear tab and column face in Figure 12b. This showed a very similar trend.

As mentioned, the UR-connection with a 22 mm thick shear tab (i.e., the 22 mm case) was tested to the maximum 4% drift and had a fracture in the base metal surrounding the bottom beam flange groove weld. The values of PEEQ index in Figures 9, 11 and 12 allowed investigating the effects of shear tabs, high-strength bolts and weld access holes in the fracture potential of the steel connections. It was found that the design demand and capacity of a shear tab can play a key factor affecting the seismic performance of the steel connection.

*4.3. Discussion*

The recently reported work [15–17] has illustrated the importance and necessity of enhancing the moment transferring from the beam webs to column faces for improving the seismic performance of steel moment connections with BWWF details. This study analyzed the demands and capacities of the shear tabs and high-strength bolts designed in accordance with the U.S. and Japan standards and compared them to the Taiwan practice. The perceptions about bolt slippage and the probably devastating effects [1,2] caused the structural engineers in Taiwan to improve the design practice. The bolted web connections were designed, using slip strength of high-strength bolts and considering the eccentricity moment. There was a small difference in the peak loads of the connections. However, the U.S. direct welded flange connection had flange buckling and strength degradation at a relatively smaller drift. The connection had a thinner shear tab and fewer high-strength

bolts. On the other hand, the other two connections had very similar design results and loading responses.

The design parameters, such as bolt arrangement and strength, shapes of welded access holes, and shear-tab thickness, were further studied with the aids of FEM simulation. Stress concentration was reported to occur in the base metal surrounding beam flange groove welds [1]. The results of parametric analysis indicated that there is a greater demand of plastic strain in the base metal of beam flanges and the surrounding welds. Moreover, the strain demand may slightly increase for fewer high-strength bolts and the lower strength. The results also agreed well with previous work [3,13,14] that the improved shapes of the weld access holes (e.g., [22,28]) may provide a smoother transition to reduce the stress concentration. It was also found that shear-tab thickness can more greatly affect the connection performance. In other words, the increase in shear-tab thickness may reduce the stress concentration and fracture potential of steel moment connections in an effective and efficient way. The findings from the above study can be considered to be of great importance not only to the retrofitting of existing SMRF buildings, but also to the design of steel moment connections with BWWF details, especially for severe seismic applications.

## 5. Conclusions

The seismic performance of a steel moment connection can vary depending on the effectiveness in transferring moment through the adopted shear tab and high-strength bolts. To investigate the effect, this study developed FEM models for two full-scale steel H-beam-to-box-column connections and validated the effectiveness by the results of seismic tests. The connections were characteristic of BWWF details. One connection tested the post-earthquake design practice adopted in the field of structural engineering in Taiwan. The perceptions about bolt slippage and the probably devastating effects have caused the structural engineers in Taiwan to improve the design practice. The bolted web connections were designed, using slip strength of high-strength bolts and considering the eccentricity moment. The other connection examined a new retrofit scheme of adding another shear tab and high-strength bolts to a pre-Northridge connection. Both the connections were tested under the 2016 AISC seismic condition and sustained a drift of 4%. The FEM models were then used to study the design demands and capacities of shear-tabs and high-strength bolts in accordance with the U.S. and Japan standards and compared to the Taiwan practice. For the ease of comparing, the von Mises stress and equivalent plastic strain (PEEQ index) were calculated in base metal surrounding the beam flange groove weld, at the root of a weld access hole and in the fillet weld between shear tab and column face. The effects were also investigated of bolt arrangement and shapes of weld access holes.

Based on the results of FEM simulation and case study, the following conclusions can be made regarding the steel connections with BWWF details:

1.  The design demands and capacity of bolted web connections were analyzed and compared. For the U.S. direct welded flange connection, the bolted web connection was designed to transfer the beam shear only. On the other hand, the perceptions about bolt slippage and the probably devastating effects caused Taiwan to improve the design practice. The slip strength of high-strength bolts was used instead, and consideration was taken of the eccentricity moment. In Japan, in contrast, the bolted web connection was designed to transfer the beam shear and to take the moment of the beam web.

2.  The design consideration and examples were studied with the aid of FEM simulation. The result indicates that there may be a small difference in the peak loads of the connections designed in accordance with the U.S., Taiwan and Japan standards. However, the U.S. direct welded flange connection can have flange buckling and strength degradation at a relatively smaller drift. The connection had a thinner shear tab and fewer high-strength bolts. On the other hand, the other two connections had very similar design results and loading responses.

3.  The design parameters of steel moment connections were further studied with the aids of FEM simulation. It was found that shear-tab thickness may affect the seismic performance of the connections more than shapes of weld access holes, and bolt arrangements and strength. The greater the shear-tab thickness, the smaller the von Mises stress and PEEQ index. This means that the increase in shear-tab thickness can reduce the stress concentration and fracture potential of steel connections in an effective and efficient way.

4.  For a new steel moment connection, it is recommended to design a shear tab with shear strength and moment capacity greater than the beam web, and to use the slip strength of high-strength bolts in the evaluation of eccentric moment in the bolted web connection. For seismic upgrading of an existing connection, it is recommended to add another shear tab and to replace with higher strength bolts. Instead of yield strength, tensile strength of steel plates should be used to evaluate the strength demand of fillet welds between shear tabs and column faces.

**Author Contributions:** C.-M.L.: Conceptualization, methodology, supervision, writing—review and editing; C.-Y.Y.: formal analysis, investigation, validation, writing—original draft; S.-Y.K.: visualization, writing—review and editing; H.-Y.C.: conceptualization, investigation, project administration, writing—review and editing. All authors have read and agreed to the published version of the manuscript.

**Funding:** This research was funded by the Ministry of Science and Technology (MOST) in Taiwan, grant number 109-2221-E-005-081-MY3.

**Acknowledgments:** The authors highly appreciate comments and suggestion made by Keh-Chyuan Tsai in National Taiwan University.

**Conflicts of Interest:** The authors declare no conflict of interest. The funders had no role in the design of the study; in the collection, analyses, or interpretation of data; in the writing of the manuscript, or in the decision to publish the results.

**Notations:**
The following symbols are used in this paper:

| | |
|---|---|
| $E$ | Young's modulus (GPa) |
| $E_{sh}$ | strain hardening rate |
| $PEEQ$ | equivalent plastic strain |
| $\varepsilon_{ij}$ | vector component of plastic strain in the direction of $i$ and $j$ |
| $\sigma_\theta = \sqrt{\frac{1}{2}[(\sigma_1 - \sigma_2)^2 + (\sigma_2 - \sigma_3)^2 + (\sigma_3 - \sigma_1)^2]}$ | von Mises stress (MPa) |
| $\sigma_1, \sigma_2, \sigma_3$ | principal stresses (MPa) |
| $v$ | Poisson's ratio |

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
