# Peer review of "Effects of Shear Tabs and High-Strength Bolts in Seismic Performance of Steel Moment Connections"

_buildings, doi:10.3390/buildings11090415_

Round 1
Reviewer 1 Report
This paper addresses a practical problem related to the role of shear tabs and high strength bolts in the seismic performance of steel moment connections. The authors used a validated finite element model to study this effect. The authors used the simulation results to propose a new design recommendation for the shear tabs in such connections. In general, the paper describes the scope of the work and techniques used very well but it suffers from some major and minor problems. The followings are the reviewer comments divided into major and minor comments. Major Comments: 1. In the introduction, I suggest that the authors add some studies related to previous work on the member level and system level to showcase the significance and uniqueness of the presented work. Some references “not all” for the authors are: Elkady and Lignos (2015), Wu et al. (2018), Sediek et al. (2020), and Cravero et al. (2020). 2. In section 2.4: provide more details about the performed calibration and show the calibration results. 3. In section 2.5: what are the end boundary conditions of the column and beam? 4. In Line 141: Is this example done by the authors? 5. In Line 186: How is the fracture modeled in the fillet weld? 6. PEEQ: I would suggest that the authors replace this term with something more meaningful to the reader (i.e. something related to the moment capacity of the connection). Right now, a researcher or an engineer cannot relate this term to design and practice. Minor Comments: 1. In the title, it should be “shear” not “sher”. 2. Line 7 in the abstract and Line 23 in the introduction, the sentence starting with “The connections were characteristics …” : revise the structure of this sentence. 3. Line 53 and Section 4.2 in the introduction: use the term “parametric” instead of “parameter”. 4. Add legend to Fig. 8. 5. I suggest renaming Section 4.2.1 to “Effect of bolt arrangement and strength” References: Cravero, J., Elkady, A., and Lignos, D. G. (2020). “Experimental Evaluation and Numerical Modeling of Wide-Flange Steel Columns Subjected to Constant and Variable Axial Load Coupled with Lateral Drift Demands,” J. Struct. Eng., 360(3), 1–19. 10.1061/(ASCE)ST.1943-541X.0002499 Elkady, A., and Lignos, D. G. (2015). “Analytical investigation of the cyclic behavior and plastic hinge formation in deep wide-flange steel beam-columns” Bull. Earthquake Eng., 1097–1118. 10.1007/s10518-014-9640-y Sediek, O. A., Wu, T.-Y, McCormick, J., and El-Tawil, S. (2020). “Collapse Behavior of Hollow Structural Section Columns under Combined Axial and Lateral Loading.” J. Struct. Eng., 10.1061/(ASCE)ST.1943-541X.0002637. Wu, T.-Y., El-Tawil, S., and McCormick, J. (2018b). “Seismic collapse response of steel moment frames with deep columns.” J. Struct. Eng., 10.1061/(ASCE)ST.1943-541X.0002150, 04018145.
Reviewer 2 Report
In the Reviewer opinion the research paper entitled “Investigation into the Role of Sher Tabs and High-strength Bolts in Seismic Performance of Steel Moment Connections” is quite good.
In this paper, finite element methods (FEM) were applied to develop the models for two full-scale steel moment connections and the effectiveness was validated by the results of seismic tests. The FEM models were then used to study the design demand and capacity of shear-tabs and high-strength bolts in accordance with the U.S. and Japan standards, and compared to the Taiwan practice. The result shows that the design demand and capacity of a shear tab can play a key factor affecting the seismic performance of a steel moment connection.
Some comments which greatly enhance the understanding of the paper and its value are presented below. Specific issues that require further consideration are:
- The title of the manuscript is matched to its content but it is too long.
- In the Reviewer’s opinion, the current state of knowledge relating to the manuscript topic has been presented, but the author's contribution and novelty are not enough emphasized.
- In the Reviewer’s opinion, the bibliography, comprising 20 references, is not representative.
- Please improve the quality of drawings.
- An analysis of the manuscript content and the References shows that the manuscript under review constitutes a summary of the Author(s) achievements in the field. However, the introduction needs more attention.
- Conclusion needs to be more revised.
- In the Reviewer’s opinion the manuscript should be published in the journal after major revision.
Reviewer 3 Report
The reviewer’s comments on the paper are as follows:
Major revisions
- Page 1, Abstract. The abstract is not written correctly. It must contain only what has been done in the paper. The first sentence is to be removed.
- Pages 1 to 2, Introduction. The novelty of the article should be described more.
- Page 4, Lines 125 to 139. Describe the modelling of contact joints in more detail.
- Page 5, Figure 2. It is incomprehensible to show a simulation of bolt slippage for a lap connection if the paper is about the column connections.
- Page 6. Explain the terms: DS-connection, UR-connection.
- Page 10, Figure 8. Mark the waveforms for monotonic and cyclic loads.
- Page 13, Figure 11. Describe the differences in the models.
Minor revisions
- Page 1, Lines 18 and 37. “finite element method (FEM)” instead of “finite element methods (FEM)”.
- Page 4, Line 112. Move to the previous page.
- No reference to [11] from the bibliography list.
- Page 15, Notation. Add units to the symbols.
- Pages 15 to 16, References. Standardise bibliographic descriptions.
Reviewer 4 Report
- How to meshthe adjacent parts of H-beam and box column, and how to set the of each element?It is suggested to supplement and explain clearly in thismanuscript
- How to consider the constraint of h-beam-to-box-column connection in the FEA calculationmodel, and if its stiffness is considered?It is suggested to supplement and explain clearly in thismanuscript.
- In the introduction, the review of existing literature is not comprehensive, can includebut not limited to, refereeingto the calculation model and stress characteristics of joint bending published by Bida Zhao, such as, https://doi.org/10.1061/(ASCE)ST.1943-541X.0002507. https://doi.org/10.1016/j.istruc.2019.10.003. https://doi.org/10.1007/s13296-018-0168-x.
Round 2
Reviewer 1 Report
I think the paper is ready to be published. I don't have any further comments.
Reviewer 2 Report
The manuscript has been sufficiently improved to warrant publication in Buildings.
Reviewer 4 Report
The authors have addressed most of the comments raised by the reviewers satisfactorily.
Now, the manuscript is well written and the topic interesting and worth of
investigation.I think it can be accepted as it is.
